# Microfluidics Facilitates the Development of Single-Cell RNA Sequencing

**DOI:** 10.3390/bios12070450

**Published:** 2022-06-24

**Authors:** Yating Pan, Wenjian Cao, Ying Mu, Qiangyuan Zhu

**Affiliations:** 1Research Center for Analytical Instrumentation, State Key Laboratory of Industrial Control Technology, College of Control Science and Engineering, Zhejiang University, Hangzhou 310027, China; 22007053@zju.edu.cn (Y.P.); wenjian.cao@zju.edu.cn (W.C.); 2College of Life Sciences, Zhejiang University, Hangzhou 310058, China; 3Huzhou Institute of Zhejiang University, Huzhou 313002, China

**Keywords:** scRNA-seq, microfluidics, droplet, microwell

## Abstract

Single-cell RNA sequencing (scRNA-seq) technology provides a powerful tool for understanding complex biosystems at the single-cell and single-molecule level. The past decade has been a golden period for the development of single-cell sequencing, with scRNA-seq undergoing a tremendous leap in sensitivity and throughput. The application of droplet- and microwell-based microfluidics in scRNA-seq has contributed greatly to improving sequencing throughput. This review introduces the history of development and important technical factors of scRNA-seq. We mainly focus on the role of microfluidics in facilitating the development of scRNA-seq technology. To end, we discuss the future directions for scRNA-seq.

## 1. Introduction

The most powerful way to study overall cell state, differentially expressed genes, gene functions and gene expression regulation mechanisms is to delineate gene expression profiles. Comprehensive gene expression profiling was first achieved by microarrays, which enabled the study of thousands of genes simultaneously. However, microarrays are limited to the analysis of known genes, and their sensitivity and dynamic range is not sufficient to meet further needs [1]. With the development of next-generation sequencing, the cost of nucleic acid sequencing has been greatly reduced, and the throughput has been improving continuously [2]. Transcriptome sequencing has gradually replaced microarrays with its obvious advantages, and has become the preferred method for transcriptome analysis [3,4]. According to the frequency of reads, the expression level of corresponding genes can be directly quantified by transcriptome sequencing [5]. More importantly, some previously unknown genes can be discovered without the need to design specific probes in advance [6,7].

The expression profile generated by traditional bulk RNA sequencing reflects the average of cell populations, and the true expression profiles of each cell is blurred, since nucleic acids need to be extracted from cultured cell populations or tissue blocks to obtain sufficient input material for sequencing [8,9]. However, it is now well established that the gene expression profiles can be very different between the same type of cells due to stochastic processes or extrinsic factors [10]. Cellular heterogeneity plays an important role in normal physiological functions and pathological changes of tissues [11,12]. For example, intratumoral heterogeneity is considered to be the main driver of tumor therapy resistance, metastasis and poor prognosis [13,14,15]. Unlike bulk RNA sequencing, to analyze transcriptome at the single-cell level can reveal the true gene expression profile of single cells, and build a deeper knowledge of the molecular mechanism [16,17]. scRNA-seq is the most powerful tool for characterizing cellular heterogeneity, which has helped us to gain a comprehensive cell atlas of organs, discover new cell types and new transcripts, identify cell-type-specific genes and cell markers, depict fine gene regulatory networks, reveal potential gene functions, reconstruct the trajectory of cell differentiation and disease occurrence, and study responses of single cells to perturbation [18,19,20,21,22,23]. However, due to the difficulty of single-cell manipulation, low nucleic acid content of single cells, and multidimensional data, scRNA-seq technology is costly and difficult to operate, which limit its extensive application [24,25]. Since the introduction of the first scRNA-seq technology, this field has developed rapidly, driven by the requirements of precision medicine and the advancement of other technologies, especially microfluidics.

Microfluidics are systems that manipulate small (10^−9^ to 10^−18^ liters) amounts of fluids in micrometer-scale channels or chambers [26]. With the advantages of simplifying complex assay protocols, drastically reducing sample volumes, reducing reagent costs, processing batches of samples in parallel and so on, it has demonstrated widespread utility in the fields of biology and medicine [27]. Microfluidics systems used in single-cell sequencing are the basis for large-scale single-cell sequencing, capable of processing and analyzing tens of thousands of single cells in parallel. Microfluidics occupies an important position in the development of scRNA-seq.

Improved scRNA-seq methods allow us to obtain higher-resolution and more comprehensive information on single-cell gene expressions and their regulations. However, to study cellular interactions beyond single-cell transcriptomes, and the regulation mechanisms behind it, researchers need to obtain the positional information of gene expression, and understand the relationships between transcripts and other molecules, such as proteins, epigenetic modifications and regulators. Therefore, spatial transcriptomics (ST) and single-cell multiple omics sequencing technologies, evolved from scRNA-seq technologies, have emerged as new scientific points of interest, and provide us with more powerful analysis tools [28,29,30,31].

In this review, we will introduce the general process of scRNA-seq library construction, and summarize the development of scRNA-seq technology, followed by the introduction of Smart-Seq, CEL-Seq and MATQ-seq chemistry in detail to illustrate the important biochemical reaction strategies. Then, we will focus on the application of microfluidic technologies, which gave birth to high-throughput sequencing methods. Finally, we share our viewpoints on the future development directions of scRNA-seq technologies from the aspects of improving the sensitivity of high-throughput scRNA-seq, joint omics analysis and spatial transcriptomics.

## 2. General Process of scRNA-seq

Although dozens of different scRNA-seq methods have been published [32,33,34,35], they all follow a similar general process, which is shown in Figure 1. In short, it includes single-cell dissociation and isolation, cell lysis and RNA release, reverse transcription (RT), second-strand cDNA synthesis, amplification, library preparation, sequencing and data analysis. In this section, we will introduce some technical points and solutions in sequencing library construction.

Specifically, there are two ways of single-cell dissociation: enzymatic and mechanical dissociation [36]. The dosage, digestion time, temperature, and other parameters of enzymatic digestion need to be adjusted according to the property of tissue to maintain the integrity and activity of cells as much as possible. Mechanical dissociation or laser capture microdissection (LCM) uses a microscope and laser beam to carefully dissect single cells from frozen tissue sections [37]. This technology was developed at the National Cancer Institute (NCI), originally for the study of heterogeneous cell populations in tumors [38]. LCM has been widely used for single-cell analysis [39,40,41,42], with the advantage of preserving the spatial information of single cells in tissues and facilitating the analysis of cell–cell and cell–environment interactions [43,44,45]. However, the main restriction that limits the application of LCM in single-cell sequencing is its low throughput. In contrast, a large number of single cells can be easily obtained by enzymatic digestion, which is suitable for high-throughput analysis. However, the spatial information of the cells is inevitably lost in the dissociation step [46]. In addition, it is difficult to obtain intact single cells from some samples (e.g., brain tissue), so it is necessary to extract a single nucleus for sequencing. At present, many sequencing technologies [47,48,49,50] have been optimized for single-nucleus sequencing.

Single-cell isolation is a crucial point in single-cell sequencing. In the early days, the common method to isolate single cells is plate-based isolation. Single cells are picked manually with microcapillary pipettes under the microscope or sorted by flow cytometry, and then distributed into individual wells of a 96/386-well capture plate, and prepared for the subsequent operations [51]. Manual cell picking is time-consuming, low-throughput, inefficient, and requires a certain degree of micromanipulation skill. Fluorescence-activated cell sorting (FACS), a widely used technique for cell sorting, can process thousands of cells in a short time, which greatly increases the throughput, but it is hard to operate for non-professionals. In addition, >10,000 cells are required as a starting input [52]; therefore, FACS should be used conservatively when analyzing rare samples. Currently, microfluidic-based single-cell manipulation methods have become the mainstream methods of cell isolation, including droplet-based methods, microwell-based methods and the commercialized Fluidigm C1 integrated fluidic circuit (IFC) system.

Due to the limited amount of nucleic acid in a single cell, RNA molecules need to be amplified after RT to meet the requirements of sequencing. The general methods of amplification include exponential amplification based on polymerase chain reaction (PCR), and linear amplification based on in vitro transcription (IVT). PCR-based amplification can easily amplify a large number of cDNAs in a short time. However, since PCR is an exponential process that creates amplification bias, the excessive amplification of some sequences, and insufficient amplification of others, will result in inaccurate transcript quantification, and amplification errors will be propagated permanently if not properly corrected [53,54]. Linear amplification is thought to be more accurate and reproducible than PCR [55]. The second method, IVT, was first introduced by Eberwine [56]. In this strategy, an upstream T7 promoter sequence, which is used to initiate IVT, is included in the RT primer. After RT and second-strand cDNA synthesis, T7 polymerase recognizes the T7 promoter and catalyzes transcription to produce more RNA molecules. Amplified RNAs need to be reconverted into DNAs for sequencing library construction. Therefore, this method is time-consuming (~28 h starting from IVT), and increases the complexity of operation, so it is not as widely used as PCR [32].

In addition to the technical difficulties in library construction, the development of analysis pipelines that are valid for scRNA-seq data analysis is also an important technical challenge. Summaries of sequencing data analysis can be obtained in other reviews [57,58,59], and are not covered here.

## 3. Developmental Course of scRNA-seq

### 3.1. Scaling of Sequencing Throughput

Since Tang et al. performed RNA sequencing in single mouse blastomeres, for the first time, in 2009 [35], many complex questions about living systems composed of cells have been answered by scRNA-seq [51,60,61]. One of the main focal points is how many types of cells are present in functional tissues. Identifying all kinds of cell types in an unbiased and accurate manner, especially those with a low proportion, requires us to analyze a large number of cells [62]. With the efforts of many researchers, scRNA-seq technology has developed from low-throughput mode to high-throughput mode. Figure 2 shows several landmark techniques that mark the development of scRNA-seq technology in terms of throughput.

The initial scRNA-seq methods can only analyze a few cells at a time, because of the time-consuming and laborious manual cell isolation and separate sequencing library construction. In order to pool all of the transcripts from different cells for library construction and sequencing, while preserving information about their cellular origin, protocols such as STRT-seq [1] and CEL-Seq [32] introduced cell-specific barcodes. All transcripts from a single cell are labeled with the same barcode, which is unique for each cell. The barcode is a short oligonucleotide sequence that can be identified through sequencing. According to the barcode information, transcripts can be easily assigned to the corresponding cells. Barcoding is an excellent strategy to achieve parallel processing. However, the introduction of barcodes alone cannot solve the difficulty of isolating a large number of cells. Later, by combining FACS with automatic liquid handling, MARS-Seq [63] successfully sequenced thousands of cells at one time. Three levels of barcodes are used to tag mRNAs, cells and plates, respectively, to pool all the materials for the subsequent automated processing. Similarly, STRT-Seq-2i [64] utilizes a specialized FACS and barcoding protocol to increase the scale of sequencing. A custom aluminum plate with 9600 wells arranged in 96 subarrays is constructed for two rounds of barcode addition, allowing 9600 cells to be sequenced in parallel. However, these plate-based methods are not easy to carry out, and the number of cells analyzed is still limited.

With the introduction of microfluidic technology, the technical barriers of high-throughput single-cell operations have been fundamentally solved. The first commercial automated microfluidic platform, the Fluidigm C1 system, enables 96 single cells to be automatically processed at one time [65]. Nonetheless, its processing capacity is far from meeting the needs of large-scale analysis. In 2015, the emergence of two droplet-based scRNA-seq technologies [33,66] was a revolutionary breakthrough in the single-cell sequencing field, enabling the simultaneous processing of thousands of cells, and truly realizing high-throughput parallel sequencing. Based on microfluidic droplets, the commercial platform 10x Genomics Chromium [67] was developed rapidly, which can characterize tens of thousands of cells at a time. The three technologies will be described in detail below.

More recently, sci-RNA-seq [68] and SPLiT-seq [69] using a combinatorial indexing strategy to label cells were successively published. Instead of physical compartmentation of single cells, the purpose of labeling more than 100,000 single-cell transcriptomes at one time can be achieved simply by multiple rounds of splitting and pooling. These methods are very easy to operate, have high cell labeling efficiencies, and can considerably reduce the cost of sequencing. Furthermore, the more cells sequenced at one time, the lower the cost of sequencing per cell.

### 3.2. Improvement in Sensitivity

In addition to the capacity to sequence multiple cells in parallel, sensitivity, accuracy, repeatability, technical noise, cost and other features are also important aspects to be considered in the design of scRNA-seq methods. Especially, sensitivity is the most critical feature, being a fundamental indicator of the performance of a method. The main events marking the development of sensitivity are shown in Figure 2.

Sensitivity can be interpreted as the probability of capturing a particular transcript and eventually detecting it by sequencing [70]. For some transcripts with low expression levels, drop-out events occur frequently in low-sensitivity sequencing platforms [71]. Low sensitivity will also reduce the accuracy and repeatability of transcript quantification, which is detrimental for the distinction of subtle differences between cell subpopulations and accurate cell type classification [72]. Every step of sequencing library construction may cause the loss of transcripts, thereby impairing the sensitivity of sequencing methods. The primary aim of all methods is to convert mRNA into cDNA for amplification. Most methods, including Smart-Seq [73] and CEL-Seq [32], use poly(T) primers to capture mRNA through the poly(A) tail to initiate RT reactions. While it can selectively capture mRNA and easily filter out numerous rRNA molecules, it also excludes some important transcripts without a poly(A) tail, such as circRNA, miRNA and nascent RNA. By contrast, MATQ-seq [74] uses random primers to capture transcripts, which can not only detect all types of transcripts, but also improve the mRNA capture efficiency, and thus achieve higher sensitivity. Another whole-transcriptome analysis method, SUPeR-seq [75], also takes advantage of random primers. After first-strand cDNA synthesis, in those methods that use PCR for cDNA amplification, the addition of the second PCR handle is also a key step in determining the efficiency of conversion. Some methods use a transferase to add a homopolymer tail, such as the poly(A) tail in SUPeR-seq or the poly(C) tail in MATQ-seq, to the 3′ end of the first-strand cDNA. A poly(T) or poly(G) primer containing the second PCR handle anneals to the homopolymer tail for second-strand synthesis. Smart-Seq uses a more convenient approach known as template-switching to incorporate another PCR handle. This method can obtain the full-length transcripts and reduce the 3′-end bias that exists in homopolymer tailing approach. However, the efficiencies of both these reactions are not 100%, and a proportion of cDNAs will inevitably be lost. To address this problem, Seq-Well S3 [76] uses random primers to initiate second-strand cDNA synthesis, and recovers most cDNA molecules without the second PCR handle. In addition, avoiding the loss of nucleic acid molecules during operation contributes substantially to the improvement in sensitivity. This is particularly evident in high-throughput methods. Cell fixation is necessary in some sequencing methods, resulting in the loss of transcripts and impaired sensitivity. For this reason, methods based on combinatorial indexing, such as sci-RNA-seq and SPLiT-seq, cannot completely replace the microfluidic-based method, despite having many benefits.

The microfluidics method reduces the reaction volume from microliters to nanoliters; thus, the sensitivity will be improved, along with an increased concentration of the targets, i.e., transcriptomes from single cells. However, when optimized, the sensitivities of the reactions in the tubes can reach the same levels as those using microfluidics. The application of microfluidics in scRNA-seq is key to improving the throughput. To improve the sensitivity, we still need to focus on the fundamental chemistry utilized in scRNA-seq methods. The key to improving the sensitivity of scRNA-seq can be split into two aspects: (1) increasing the capture efficiency of RNA during the first-strand synthesis; (2) increasing the efficiency of the conversion of cDNA into amplifiable products, regardless of using second-strand synthesis or the template-switching activity of RT enzymes. Therefore, by integrating scRNA-seq chemistry with microfluidics, a higher sensitivity could be more easily achieved; meanwhile, the target concentration is greatly increased and background signal is reduced. Other smart strategies improving accuracy, reducing cost and so on will be discussed in the introduction of specific methods.

## 4. Low-Throughput scRNA-seq Methods

The chemistry developed in low-throughput methods is also fundamental to high-throughput methods. We will focus on the critical chemistries of three classic low-throughput methods to pave the way for the introduction to high-throughput methods.

### 4.1. Smart-Seq Chemistry

Smart-Seq [73] is the most popular method, and is still considered the “gold standard” of scRNA-seq. The key of the Smart-Seq technique is to use the template-switching mechanism at the 5′ end of the RNA transcript (SMART) to convert mRNA into an amplifiable cDNA sequence. The Moloney murine leukemia (M-MLV) reverse transcriptase used in Smart-Seq for RT reactions has the characteristics of automatically adding three additional nucleotides (usually three cytosines) at the end of the cDNA, and inducing the activity of template-switching. After first-strand synthesis, template-switching is performed using a template-switching oligo (TSO, carrying three riboguanosines at its 3′ end) to add the PCR adaptor to the 3′ end of cDNA. The resulting cDNA molecules can be amplified by PCR. All of the above reactions are completed in one step.

In the following years, Rickard Sandberg’s group continued to improve this technology, and developed Smart-Seq2 [10] and Smart-Seq3 [77] to further improve the sensitivity, accuracy and so on. In Smart-Seq2, the buffer composition, reaction temperature and TSO sequence are improved to obtain higher sensitivity, fewer technical biases and less variability. The specific scheme is not discussed in this review, but can be found in the Smart-Seq2 protocol [10].

In Smart-Seq3, the sensitivity is further improved by optimizing the enzymes and buffers used in RT and preamplification reactions. More importantly, this technique combines full-length transcriptome coverage with a unique molecular identifier (UMI) counting strategy to improve the accuracy of transcript counting without loss of coverage. UMIs are extensive synthetic random sequences. There are enough different sequences to ensure that all transcripts in a single cell can be tagged with unique tags. After sequencing, the number of original transcripts can be counted accurately by UMIs to eliminate the bias of PCR amplification [78]. At present, almost all scRNA-seq techniques use the second-generation short-read sequencing platform; that is, long sequences need to be sheared into short fragments for sequencing. For most of the sequencing library construction methods using barcodes and UMI tags, the effective cDNA sequences that can be used for analysis are only the 3′ or 5′ ends of the complete cDNA sequences, because barcodes and UMI tags can only be added to the ends of the cDNA. When a large amount of sequence information in the middle of the transcript is lost, the nucleotide and splicing variation information of the transcript is unobtainable. On the contrary, Smart-Seq and Smart-Seq2 do not use barcodes or UMIs to label transcripts, so full-length cDNA can be recovered. One of the consequent disadvantages is that not all fragments analyzed by sequencing carry cell barcodes, which means that this method is incompatible with high-throughput, parallel single-cell sequencing. Secondly, the amplification bias caused by PCR cannot be corrected without a UMI tag. Smart-Seq3 addressed the incompatibility between UMI and full-length transcript coverage. The TSO primer used in Smart-Seq3 contains an 11-bp tag sequence and an 8-bp UMI sequence, as well as three riboguanosines. After sequencing, 5′ UMI-labeled reads with different lengths and a group of internal reads without UMIs are obtained. The 11-bp tag is used to unambiguously distinguish 5′ UMI reads from internal reads. 5′ UMI reads are used to accurately count the number of transcripts, and the internal reads are used to reconstruct full-length transcripts. 

### 4.2. CEL-Seq Chemistry

The sequencing library construction strategy of CEL-Seq [32] is significantly different from that of Smart-Seq. In CEL-Seq, IVT is used for cDNA amplification. The RT primer contains poly(T), a cell-specific barcode, 5′ Illumina sequencing adaptor and T7 promoter. Since barcode sequences are introduced in the RT step, cDNA molecules in different cells can be pooled together for IVT after RT. The IVT reaction requires 400 pg total RNA as the minimum input material for a single round of amplification, and the pooling of cDNAs in several cells can meet the material requirement of IVT. Then, driven by the T7 promoter, cDNAs can be linearly amplified by IVT. Then, these amplified RNAs are converted to cDNAs for sequencing. In CEL-Seq2 [79], UMI is added to the RT primer to accurately count transcripts.

### 4.3. MATQ-seq Chemistry

At present, the most sensitive sequencing method is MATQ-seq [74], published by Zong’s group in 2017. 

They used random primers developed from multiple annealing and looping-based amplification cycles (MALBAC) [80] to capture RNAs. Combined with ten cycles of annealing, full-length RNA can be converted into cDNA with high efficiency. In addition, non-polyadenylated RNA can also be captured. After RT, a poly(C) tail is added to the 3′ end of cDNA for second-strand synthesis. UMI is introduced during second-strand synthesis to reduce the bias caused by subsequent PCR. MATQ-seq achieved 89.2 ± 13.2% capture efficiency, and greatly increased the detection efficiency for low-abundance genes. The high sensitivity and accuracy of MATQ-seq enables the detection of subtle differences in gene expression between single cells from the same population.

## 5. High-Throughput scRNA-seq Methods

### 5.1. Application of Microfluidic Technology in scRNA-seq

Small sample volume and a large number of samples are two evident characteristics of single-cell operation. Microfluidics, as a technology for manipulating liquid at the nanoliter level and in a multiplexing manner, has inherent advantages in single-cell research. Microfluidics, also known as lab-on-a-chip, is to perform sample preparation, reaction, separation, detection and other basic operations involved in the fields of chemistry and biology on a single chip of several square centimeters, or even smaller [81,82]. 

Beyond chemistry, single-cell manipulation is a main technical challenge of single-cell sequencing. Microfluidics enables parallel manipulation of single cells in a high-throughput manner. In scRNA-seq, it is primarily used to physically isolate individual cells.

The first widely used commercial microfluidic system for scRNA-seq is the Fluidigm C1 system, developed in 2012. The core of this platform is an integrated fluidic chip, which can isolate 96 cells at a time. After isolation, the number and quality of captured cells can be assessed using a microscope. Then, cells are sent to subsequent processing stage in an automated and parallelized manner. The disadvantages of this system are obvious. To begin with, it requires at least 10,000 cells as input, and the number of cells it can handle is very limited, suggesting that it is very inefficient and unsuitable for analyzing rare cell populations [60]. Additionally, the C1 system performs badly on heterogeneous cell populations due to it requirements for cell size and shape. [65]. A specific model of C1 chip can only capture cells of a certain size. Small cells are difficult to confine to the capture site, resulting in low capture efficiency. On the other hand, large cells are prone to becoming stuck. The C1 system assumes that cells are in a rounded shape, so some types of cells, such as adult cardiomyocytes, whose shapes are irregular, are incompatible with this platform [65]. Furthermore, the throughput of this system is far from meeting the needs of large-scale sample analysis. An scRNA-seq study in mouse pancreatic islet cells by Xin et al. using the C1 system showed that a large number of cells are damaged or fuse with other cells during the capture process, resulting in severe cross-contamination and markedly altering gene expression patterns [83]. 

In the laboratory, two kinds of integrated microfluidic systems are designed for single-cell isolation. The original design for physically isolating cells is to trap them in droplets using droplet microfluidic chips. There are three widely used geometric structures of droplet microfluidic channels: T-junctions, flow-focusing and co-flow (Figure 3a–c). By connecting multiple droplet generators in tandem, we can obtain oil-in-water, water-in-oil and/or multiple droplets (O/W/O, W/O/W, etc.). Here, we illustrate the principle of droplet generation, taking the most commonly used flow-focusing channel for the generation of water-in-oil droplets as an example. The flow-focusing geometry consists of two vertically crossing channels. The oil phase flows in from two side channels perpendicular to the water phase, and converges into the channel of the water phase. When the shear force or stress of the oil phase reaches a certain value, the water phase can form small emulsions (10^−15^~10^−9^ L), which are carried by the oil [84]. As long as the flow rates of the two phases remain stable, droplets can be generated continuously and stably at a high frequency. In single-cell sequencing experiments, a single-cell suspension is loaded as an aqueous phase after diluting it to an appropriate concentration according to the principle of Poisson distribution. A droplet containing one cell can be regarded as an independent reactor for subsequent experiments. 

Another strategy is to use a microwell array as a single-cell container (Figure 3d). Tens of thousands of nanoliter or picoliter microwells are fabricated in an area of several square centimeters. Each individual cell is loaded into a single well, which is used as an independent reactor for cell lysis, RT and barcode labeling. 

In the following, we will discuss and compare the scRNA-seq methods based on microfluidics in detail.

### 5.2. Droplet-Based scRNA-seq Methods

#### 5.2.1. inDrop

In 2015, Klein et. al. [66] and Macosko et. al. [33] each published a droplet-based scRNA-seq technology, respectively, in *Cell*. The publication of these two technologies is another milestone in this field.

Figure 4a(ⅰ) shows the inDrop workflow. The core of the inDrop [66] method is to co-encapsulate the cell and the barcoded hydrogel microsphere (BHM) to produce cell-specific barcodes. BHMs are generated using droplet microfluidic devices. An acrylamide: bis-acrylamide solution is used as the aqueous phase to generate droplets, which are then polymerized into tangible microspheres in the presence of APS (ammonium persulphate) and TEMED (N, N, N′, N′-tetramethylethylenediamine). Acrydite-modified DNA primers are added to the prepolymer and incorporated into the hydrogel mesh upon acrylamide polymerization. Then, they use a split-pool combinatorial barcoding strategy to synthesis unique barcodes from BHMs through two extension steps. Each BHM carries one of the 147,456 unique barcodes, and this BHM library can label 3000 cells in one reaction with a 99% unique labeling ratio. It is worth noting that the 5′ end of the DNA primer connected to the bead has a photo-cleavable spacer. After the co-encapsulation of the cell and BHM, the spacer is cleaved under UV irradiation, and the primer is released and anneals with RNA to start RT reactions. This is the key to improving RNA capture efficiency, because, compared with most other bead-based barcoded primer delivery methods, the released primers can bind RNA molecules more effectively. 

A droplet microfluidic device was designed to realize the co-encapsulation of a single cell and BHM. The structure is shown in Figure 4a(ⅱ). Cells, BHMs and RT/lysis reagents are pumped into each inlet, respectively, and co-emulsified into droplets by carrier oil. The size of the channel is well-designed to ensure that the BHMs are closely packed and regularly released (Figure 4a(ⅲ)). When BHM release synchronizes with droplet formation, 100% hydrogel packaging efficiency can be achieved. By contrast, cell loading follows the Poisson distribution. Through the limited dilution of single-cell suspension, the probability of two cells being packed into the same droplet can be low.

Although this method realizes large-scale parallel RNA sequencing, the capture efficiency of transcripts is relatively low, and genes with transcriptional abundance less than 20–50 transcripts cannot be reliably detected in a single cell.

#### 5.2.2. Drop-Seq

The twin of inDrop, Drop-Seq [33], shares the same strategy with inDrop in terms of barcoding transcripts (Figure 4b). The two technologies differ in the following aspects:

First, the material of the beads used in Drop-Seq is a type of hard resin that is not deformable. Both beads and cells obey the Poisson distribution, resulting in a lower package efficiency. Only 1% of droplets are the desired droplets, which contain one cell and one barcoded bead. Barcoded primers are synthesized directly onto the resin beads. Twelve split-pool cycles of oligonucleotide synthesis are used to synthesize barcode sequences, resulting in 4^12^ (16,777,216) possible unique sequences.

Secondly, the primers cannot be released from the beads. After RT, all cDNAs are covalently connected to beads, which are called “single-cell transcriptomes attached to microparticles” (STAMPs). This might be another drawback of Drop-Seq compared to inDrop, which we have mentioned above. 

Thirdly, in Drop-Seq, droplets are broken immediately after the annealing of primers and mRNAs, and all STAMPs are pooled together for RT reactions. By contrast, in inDrop protocol, RT reagents are co-encapsulated into droplets, and RT reactions are performed individually in each droplet. Notably, bulk RT reactions can prevent the risk of RT inhibition in picoliter volume reactions and reduce batch effect [36].

Finally, after obtaining cDNA molecules, Drop-Seq uses the Smart-Seq library construction method to build a sequencing library, while inDrop uses the CEL-Seq method. Therefore, they inherit the advantages and disadvantages of Smart-Seq and CEL-Seq, respectively.

#### 5.2.3. 10x Genomics

At present, the commercial 10x Genomics Chromium platform has the best comprehensive performance among the high-throughput scRNA-seq platforms [67].

This platform is also droplet-based, and has some similar characteristics to Drop-Seq and inDrop. The overall workflow of 10x Chromium is shown in Figure 4c. The core of this technology is gel beads in emulsion (GEM). An eight-channel microfluidic chip is used to generate GEM. In each microfluidic channel, about 100,000 GEMs are generated every 6 min, encapsulating thousands of cells in GEMs. Short droplet generation times can minimize the transcriptional perturbation. Gel beads are encapsulated by the inDrop method, and the encapsulation efficiency can reach about 80%. Gel beads are functionalized with barcoded primers. Once the beads are encapsulated into droplets, these primers are released as the beads dissolve. Interestingly, this is considered to be the key to the high sensitivity of 10x Genomics Chromium. The cell capture efficiency can also reach 50%, which gives this system the advantage of analyzing rare samples.

#### 5.2.4. BAG-seq

All of the above droplet-based methods basically use the strategy of co-encapsulation of cells and barcoded beads to deliver cell-specific barcodes during RT. Recently, Li et al. [85] reported a completely different strategy utilizing cell barcodes (Figure 4d).

They used the droplet generation microfluidic chip in DroNc [47] (a modified version of Drop-Seq for single-cell nuclear sequencing) to package single cells. One aqueous phase is cell suspension, and the other aqueous phase consists of reagents containing polyacrylamide prepolymer, 5′-acrydite-modified primers and detergents. In addition, the oil phase contains TEMED, which can accelerate the polymerization of polyacrylamide hydrogel. Once the droplets are formed, they begin to polymerize under the action of TEMED, and the 5′-acrydite-modified primers co-polymerize into the gel network. The cells are lysed by the detergents, and the released mRNAs are captured by the covalently bonded primers, and restricted to the ball of acrylamide gel (BAG). After polymerization, each BAG is treated as an independent reaction vessel. The size of the gel mesh can allow enzymes, oligonucleotides and other reagents to diffuse freely for the subsequent reactions. Following RT, split-pool is performed to add barcodes to transcripts in each BAG.

One of the advantages of BAG-seq is that it does not rely on two simultaneous Poisson events when packaging cells, because the oligonucleotide primers used to capture mRNAs are not bound to the beads, but directly added to the reagents. Therefore, compared with other co-encapsulation methods, BAG-seq has a higher cell capture efficiency, which is particularly important for the analysis of rare and precious samples. In addition, using free primers to capture mRNAs is equivalent to increasing the volume of capture beads, which can theoretically improve the capture efficiency. Li et al. [85] have demonstrated that BAG-seq can detect more genes and unique templates than other high-throughput methods at the same sequencing depth. Furthermore, BAG-seq adds barcodes directly to cDNAs by splitting and pooling the BAGs, rather than synthesizing barcoded beads in advance. Therefore, the cost is greatly reduced. Finally, this platform can also be used for single-cell whole-genome sequencing, which demonstrates its flexibility.

### 5.3. Microwell-Based scRNA-seq Methods

#### 5.3.1. CytoSeq

CytoSeq [86], published in 2015, proved that microwells can also be used for cell isolation and barcoding. A schematic of the workflow of CytoSeq is shown in Figure 5a. At first, cells are loaded into the picoliter microwell array at a concentration that ensures monodispersity, and are then simply settled into wells by gravity. Next, the barcoded beads are loaded into the array. The optimized sizes of the wells and beads prevent two beads being loaded into one well. The beads are magnetic, so after mRNA capture, all beads can be retrieved conveniently from the array by magnet. Then, the beads can be processed together in a tube. In this protocol, instead of amplifying and sequencing the whole transcriptome, they sequenced a set of selected genes by multiplex PCR.

This protocol uses an unsealed microwell array, which significantly limits mRNA capture efficiency and increases the chance of cross-contamination.

#### 5.3.2. RNA Printing

In 2015, Bose et al. [87] reported a method that used microwell chips to print RNAs on glass, or capture RNAs on beads, to achieve solid-phase capture of RNAs. The chip consists of a microwell array and a flow channel connecting all wells. Reagents are injected into the channel and distributed into each well. The material of the chip is hydrophobic and deformable polydimethylsiloxane (PDMS), which can facilitate reversible sealing of wells by mechanical squeezing or an oil layer.

In RNA printing mode (Figure 5b(ⅰ)), cells are first loaded into the wells by gravity. After adding the lysis buffer, the microwell array is sealed immediately on the glass surface by mechanical sealer prior to cell lysis. The glass surface is covalently functionalized with oligo (dT) primers, and mRNAs released from cells are captured by these primers and “printed” on the glass surface. After mRNA immobilization, the sealer is released, and the subsequent treatments can be carried out directly on the chip.

Another mode is bead-based mRNA capture (Figure 5b(ⅱ)). Barcoded mRNA capture beads with 1 of 960 possible barcode sequences are synthesized by the split-pool strategy. After cell loading, beads are also loaded by gravity settlement. Cell suspensions loaded into the microwell can be agitated with laminar flow by reversing the flow direction repeatedly to achieve >50% cell capture efficiency. Oil is injected into the channel to seal the microwell array after adding lysis buffer. The mRNAs released from cells can hybridize to the primers on the beads. Once the mRNAs are immobilized, the oil can be removed. After washing, RT, cDNA second-strand synthesis and IVT (for preamplification) can be performed on the chip.

Solid-phase capture of RNAs can facilitate reagent exchange, removal of contaminants and imaging, which is one of the advantages of this system. More importantly, the microwell array can be reversibly sealed during cell lysis and RNA capture, which greatly improves the RNA capture efficiency and reduces the risk of cross-contamination. However, the complex manual or computer-based control of temperature, pressure and sample loading greatly limits its application.

#### 5.3.3. Seq-Well

Due to the cross-contamination, low capture efficiency, poor sealing effect and operation complexity of most microwell-based techniques, in 2017, Gierahn et al. published Seq-Well [89], which addressed these issues to some extent.

Seq-Well confines cells and barcoded beads in sub-nanoliter wells of PDMS chips to label single cells. The loading efficiency can reach ~95% for beads and 80% for cells. The operation and equipment of Seq-Well are very simple. Experiments can be completed with only a few PDMS chips, sealing membranes, pipettes, manual clamps, an oven and a tube rotator.

More importantly, Seq-Well uses a semi-permeable polycarbonate membrane (10 nm pore size) and surface-functionalized PDMS array to limit the nonspecific binding of RNA to the array, and to increase the sealing efficiency. In short, the array contains a unique three-layer surface functionalization. The surface of the nanowells is covered by poly(glutamate), while the top surface of the array is covered by chitosan. Poly(glutamate) coating can prevent the nonspecific binding of RNA, which helps reduce the loss of RNA. Chitosan coating can facilitate reversible attachment of the sealing membrane, enabling rapid solution exchange, improving the efficiency of cell lysis and transcript capture, and reducing cross-contamination.

Given the inefficiency of the template-switching strategy used in Drop-Seq for second-strand cDNA synthesis, more recently, this group published an updated method named Seq-Well S3, which incorporates a randomly primed second-strand synthesis [76]. This mode of second-strand synthesis recovers most of the RNA molecules that are reverse-transcribed into cDNAs, but failed to be tagged with the secondary PCR handle. This method has been proved to be more sensitive than Seq-Well, with sixfold gene detection sensitivity and tenfold UMI detection sensitivity.

#### 5.3.4. Microwell-seq

Another microwell-based technique, known as Microwell-seq [88], uses similar barcoded beads to capture mRNA (Figure 5d). Compared with other methods, its advantage is that single cells are captured in an agarose microarray constructed from a PDMS microcolumn array mold. The agarose microarray is easy and quick to manufacture, resulting in substantial reduction in cost. In addition, the magnetic beads can be collected conveniently and effectively. The cost of sequencing library construction for each cell is as low as USD 0.02. These authors used Microwell-seq to analyze more than 400,000 cells, covering all of the major mouse organs to construct a complete mouse cell atlas.

### 5.4. Comparison of Droplet- and Microwell-Based Methods

In this subsection, we summarize the advantages and disadvantages of the different methods to provide the reader with some guidance for choosing the appropriate method for experiments. The characteristics of each method are summarized in detail in Table 1.

Here, we mainly discuss the differences between droplet-based and microwell-based methods. Among these two kinds of methods, droplet-based methods were proposed earlier and are used more widely due to several advantages. First, theoretically, there is no upper limit on the number of droplets that can be generated in one experiment. That is, the number of cells that can be detected is scalable. The ability of the sequencing method to detect a large number of cells is necessary for identifying rare cell types and drawing detailed cell maps. Second, the generation and operation of droplets is very flexible. It is easy to fuse, split, sort and reload droplets through channel design, and to control their size and generation rate through flow rate control.

However, since a single cell and a single-barcoded bead need to be co-encapsulated into a droplet, the encapsulation efficiencies might be very low, despite some studies having proposed strategies to realize a sub-Poisson distribution. Additionally, the generation process of the droplets might lead to asynchronous cell lysis, cell expression profile disturbance and, thus, technical noise.

By comparison, microwell-based techniques are more efficient in co-encapsulation. Moreover, the loading time of cells is shorter, which reduces the differences in lysis times among cells. In addition, microwell chips are easier to operate and require less complicated peripheral equipment. Although microwell-based techniques are simpler and portable, there are two main drawbacks that make them less popular than droplet-based techniques. Firstly, the chip needs to be redesigned and refabricated when the experimental conditions, such as cell number and bead size, change. Secondly, the low RNA capture efficiency and cross-contamination issues remain poorly addressed.

## 6. Future Perspectives

### 6.1. Improvement in Sensitivity of High-Throughput Sequencing Methods

Due to the ingenuity and efforts of researchers, single-cell transcriptome sequencing technology has developed rapidly. So far, dozens of different sequencing methods have been reported. In some comparative studies [70,90,91,92], the authors have systematically compared the performance of these sequencing technologies. Interestingly, the sensitivity and accuracy of high-throughput sequencing technologies are generally lower than those of low-throughput technologies. So far, 10x Chromium, the most sensitive high-throughput sequencing technology, can only detect ~7000 genes per cell; meanwhile, MATQ-seq, the most sensitive low-throughput technology, can detect about 18,000 genes, more than twice that of 10x Chromium. An insensitive high-throughput scRNA-seq method yields unreliable data, and loses a lot of useful information. It is still one of the challenges in this field, to improve the sensitivity of high-throughput methods to be close to the level of low-throughput methods.

### 6.2. Development of Multiomics Joint Sequencing Methods

In the era of precision medicine, a comprehensive understanding of the molecular mechanisms in normal and disease processes has become an urgent demand for both basic and clinical research. Some large international research projects, such as the Human Tumor Atlas Network (HTAN) project (https://ccr.cancer.gov/research/cancer-moonshot, accessed on 1 September 2018), are not satisfied with the analysis of life systems in a single dimension, they require analysis of the deeper machinery in multiple dimensions across space and time [93]. As a result, multiomics analyses have emerged as a critical impetus for advancing precision medicine. One benefit from the rapid development of single-cell transcriptomics and other omics methods, joint profiling of multiomics has been able to perform in the same cell. Single-cell multiomics analysis is the most powerful strategy to reflect the real interaction among RNA, DNA and proteins, and to reveal regulatory networks. 

Some joint sequencing methods, which integrate transcriptomics with genomics, epigenomics, proteomics and other omics methods, have been developed to analyze molecular regulatory mechanisms in single cells. G&T-seq integrates transcriptome and genome sequencing [94] to explore the relationship between genomic changes and gene expression levels. In 2019, Ren Bing’s group developed Paired-seq [95], a technology that combines single-cell chromosome accessibility analysis and transcriptome sequencing. Integrative analysis of chromosome accessibility and gene expression profile enables deconvolution of the regulatory relationship between cis-regulatory elements and their putative target genes at the single-cell level. This group further developed Paired-Tag [96], a high-throughput sequencing method for joint profiling of histone modifications and the transcriptome in single cells to reveal the epigenetic regulatory mechanisms of cell-type-specific gene expression. Another group reported a method called scM&T-seq [97] for single-cell methylome and transcriptome sequencing. By performing it on mouse embryonic stem cells, they revealed novel associations between heterogeneously methylated distal regulatory elements and the transcription of key pluripotency genes. Fan et al. analyzed the DNA methylome, chromatin accessibility and the transcriptome simultaneously in pancreatic ductal adenocarcinoma cells [98]. The relationships among methylation, chromatin accessibility and gene expression in tumor cells have been revealed, and a set of novel candidate biomarkers for prognosis have been identified. 

In conclusion, multiomics comprehensive analyses provide a large amount of gene regulation information that single-omics methods are not able to cover. Although many multiomics sequencing technologies have been reported, the development of this field is still in its infancy. The technology that has higher throughput and can analyze more omics information at the same time, and the bioinformatic pipeline for large amounts of data analysis, need to be developed.

### 6.3. Acquisition of Spatial Information of Transcripts

Joint analysis of single-cell transcriptome and other omics data in the same cell has increasingly become the future development trend of single-cell transcriptome sequencing technology. In addition, preserving spatial information is also a target of current efforts. In a multicellular system, cells do not play a role in isolation, but interact and coordinate with each other and their surrounding environment. For example, during development, the direction of cell differentiation is not only determined by its own genetic program, but is also affected by its location and other cells. In tumors, the tumor microenvironment formed by microglia, macrophages, lymphocytes, endothelial cells and other cells has a certain form of spatial organization, and this affects the biochemical changes in the tumor and the response to treatment [99]. However, in most of the scRNA-seq protocol, tissues need to be dissociated before sequencing, resulting in the loss of spatial information. Spatial transcriptomics makes up for this deficiency, and can provide us with a large amount of information regarding the spatial gene expression heterogeneity.

The initial work to capture spatial Information was described by Nichterwitz et al., and was named LCM-seq [41]. The LCM method, which we have mentioned above, is used to quickly cut target single cells from frozen tissues. It is suitable for analyzing rare cell populations with known locations, and cell types that are difficult to isolate, such as neurons. However, low sensitivity and throughput greatly limit its application. 

Other studies have focused on creating a spatial barcode array to measure the spatial distribution of transcripts in tissue sections. A spatial barcode array is composed of a large number of information spots that contain unique positional barcodes encoded by oligonucleotides. Barcodes can be transferred to tissue sections in parallel to achieve spatial transcription mapping. After tissue section digestion, all transcripts are pooled and processed in the same manner as in scRNA-seq. This strategy was first demonstrated by Ståhl et al. [100]. However, in this work, the diameter of one spot is 100 μm, with a center-to-center distance of 200 μm. Thus, several cells can receive the same positional barcode, which means that the resolution of this method fails to reach the single-cell level. Several subsequent studies have been devoted to improving the spatial resolution. The Visium system of 10× Genomics improved the resolution to 55 μm by reducing the spot size. Two similar studies, describing Slide-seq [101] and the high-definition spatial transcriptome [102], further pushed the resolution down to 10 μm and 2 μm, respectively. The sophisticated decoding process and low sensitivity are the major limitations of these methods. 

Microfluidics also contributes towards the development of ST. More recently, Rong Fan’s team reported an easily operated method called dBiT-seq [103], based on microfluidic technology. Two microfluidic chips with 50 parallel channels are successively placed on the tissue section perpendicular to each other to deliver barcodes to transcripts. A unique barcode is introduced into each channel to label mRNA. After two rounds of labeling, cells at the intersection of two channels receive a distinct combination of barcodes. This method can reach a spatial resolution of 10 µm, with higher sensitivity than all previous methods. However, like other ST methods, dBiT-seq is not able to obtain spatial gene expression patterns at the single-cell level. Some studies have tried to integrate ST data with scRNA-seq data to identify the major cell types in each spatial spot to address this gap [103,104].

Overall, to achieve single-cell resolution is still a major technical challenge in spatial transcriptome sequencing technology at present, and more solutions need to be proposed.

An overall blueprint for the future orientation of scRNA-seq is shown in Figure 6.

## 7. Conclusions

Single-cell RNA sequencing technology has come of age in the last 13 years. Many strategies have been undertaken to resolve the technical challenges and contribute to the improved sequencing data quality and increased throughput. With the development of omics techniques and precision medicine, it has become an urgent requirement to obtain a more comprehensive and refined single-cell molecular atlas. We have been able to analyze single-cell transcriptomes in a fast and high-throughput manner with the help of microfluidics. However, a high-throughput scRNA-seq method, that also exhibits high sensitivity and accuracy, is still needed. The sensitivity of single-cell RNA sequencing is the basis for revealing accurate and comprehensive biological information, and for ensuring the data quality of multiomics sequencing and spatial transcriptomics. In addition, to ensure that scRNA-seq becomes a routine analysis method, rather than an exclusive technology among a small group of laboratories, it is necessary to further simplify the operation steps and reduce its cost. Various omics technologies at the single-cell level have also been developed, and molecular information at the single-omics level has been revealed. The acquisition of multiple omics-level information for a single timepoint and spatial location, and information on molecules for analysis of genetic spatiotemporal regulation networks, are other important directions for scRNA-seq in the future. 

## Figures and Tables

**Figure 1 biosensors-12-00450-f001:**
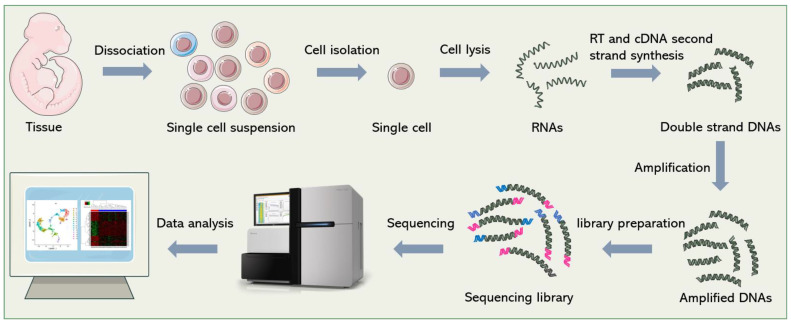
General workflow of scRNA-seq. Firstly, single-cell suspensions are obtained from cultured cells or tissue blocks. Then, cells are isolated and lysed. The released RNAs are reverse-transcribed into cDNAs. After second-strand synthesis and amplification, sequencing adapters are added to both ends of the cDNAs to construct the final sequencing library. Finally, bioinformatics pipelines are used to analyze the sequencing data and re-establish gene expression signatures of single cells.

**Figure 2 biosensors-12-00450-f002:**
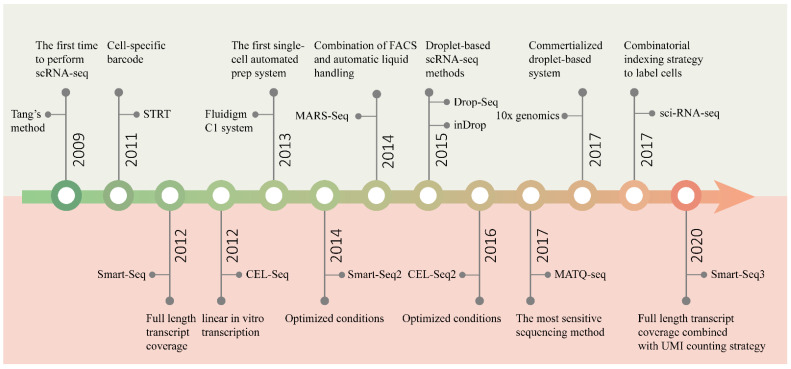
Timeline of the development of scRNA-seq technology. The upper half of the graph shows the main events marking the development of sequencing throughput, and the bottom half the improvement in sensitivity.

**Figure 3 biosensors-12-00450-f003:**
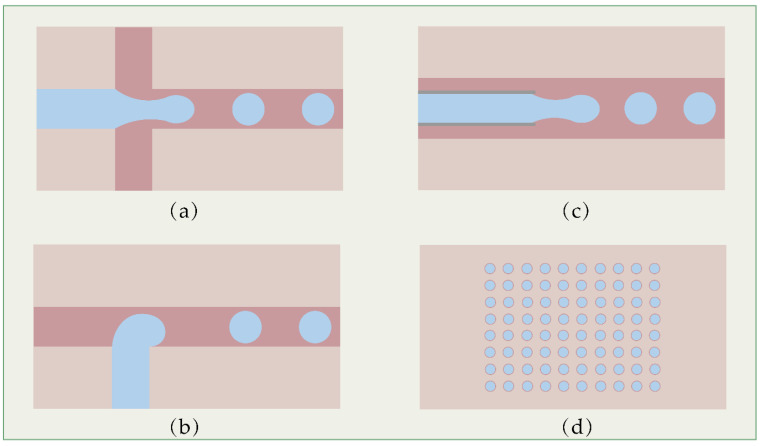
Basic structure of droplet and microwell microfluidic devices for scRNA-seq: (**a**) T-junction; (**b**) flow-focusing; (**c**) co-flow; (**d**) microwell.

**Figure 4 biosensors-12-00450-f004:**
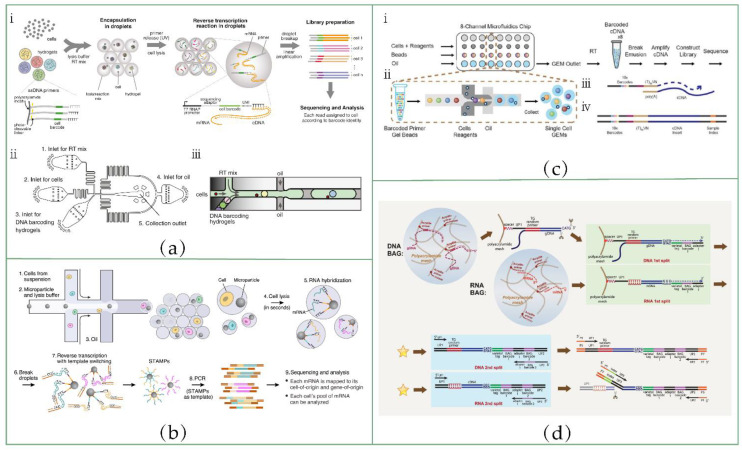
Application of droplet microfluidics in scRNA-seq: (**a**) ⅰ, inDrop RNA sequencing workflow; ⅱ, design of the microfluidic device for co-encapsulation of cells and BHM; ⅲ, schematic diagram of the process of co-encapsulation of cells and BHM. Reprinted from ref. [66]. (**b**) Schematic of sequencing library preparation with Drop-Seq. Reprinted from ref. [33]. (**c**) ⅰ, scRNA-seq workflow on 10x genomics platform; ⅱ, cells and gel beads are co-encapsulated in microfluidic chip; ⅲ, barcoded primer annealing with RNA; ⅳ, finished library molecule. Reprinted from ref. [67]. (**d**) BAG-seq workflow. Reprinted from ref. [85].

**Figure 5 biosensors-12-00450-f005:**
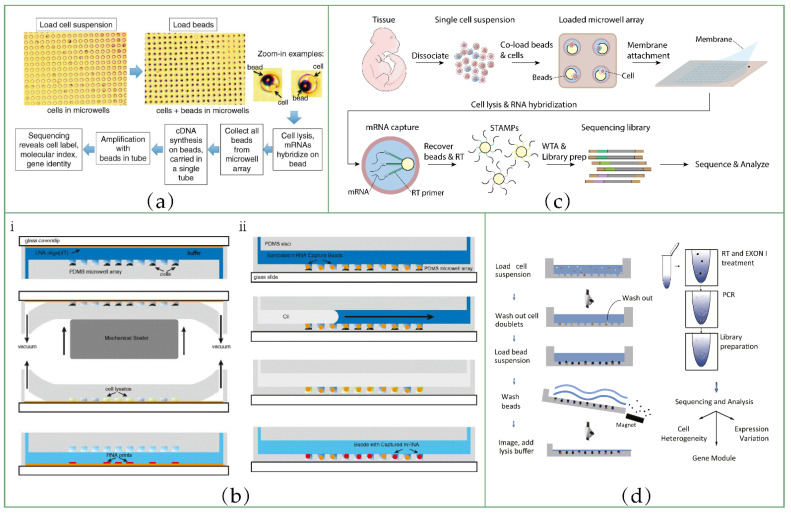
Application of microwell microfluidics in scRNA-seq: Schematic of the workflow of (**a**) CytoSeq. Reprinted from ref. [86]. (**b**) ⅰ, single-cell RNA printing; ⅱ, single-cell RNA capture on beads. Reprinted from ref. [87]. (**c**) Seq-Well and (**d**) Microwell-seq. Reprinted from ref. [88].

**Figure 6 biosensors-12-00450-f006:**
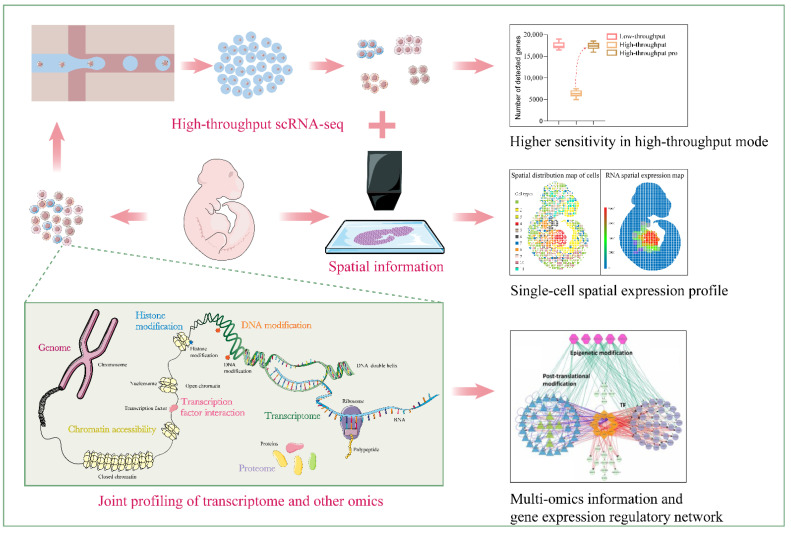
Future outlook of scRNA-seq technology. Gene expression regulatory network at the lower right corner is cited from the article of Guan D, et al. 2014. Reprinted from ref. [105].

**Table 1 biosensors-12-00450-t001:** A summary of the various scRNA-seq methods mentioned in this review.

Field	Publication Year	Name	Barcode	UMI	Amplification Method	Sequencing Method	Throughput *	Conclusion	Reference
Low-throughput methods	2009	Tang’s method	no	no	PCR	Nearly full length	+	The first scRNA-seq method	Tang F, et al. *Nat Methods*. 2009 [35]
	2011	STRT-seq	yes	no	PCR	5′ sequencing	+	1. Being able to analyze transcription start sites 2. Cell-specific barcode	Islam S, et al. *Genome Res*. 2011 [1]
	2012	Smart-seq	no	no	PCR	full length	+	1. High sensitivity 2. High coverage 3. Template-switching strategy	Ramsköld D, et al. *Nat Biotechnol*. 2012 [73]
	2012	CEL-seq	yes	no	IVT	3′ sequencing	+	Linear in vitro transcription	Hashimshony T, et al. *Cell Rep*. 2012 [32]
	2014	Smart-seq2	no	no	PCR	full length	+	Optimized conditions	Picelli S, et al. *Nat Protoc*. 2014 [10]
	2016	CEL-seq2	yes	yes	IVT	3′ sequencing	+	Optimized conditions	Hashimshony T, et al. *Genome Biol*. 2016 [79]
	2017	MATQ-seq	no	yes	Multiple annealing	full length	+	The most sensitive scRNA-seq method	Sheng K, et al. *Nat Methods*. 2017 [74]
	2020	Smart-seq3	no	yes	PCR	full length	+	Highly sensitive and isoform-specific	Hagemann-Jensen M, et al. *Nat Biotechnol*. 2020 [77]
Automatic liquid handling high-throughput method	2014	MARS-Seq	yes	yes	IVT	3′ sequencing	++	Combination of FACS and automatic liquid handling	Jaitin DA, et al. *Science*. 2014 [63]
Droplet-based high-throughput methods	2015	inDrop	yes	yes	IVT	3′ sequencing	++	1. High hydrogel packaging efficiency 2. UV-initiated primer release 3. High-throughput CEL-seq method	Klein AM, et al. *Cell*. 2015 [66]
	2015	Drop-seq	yes	yes	PCR	3′ sequencing	++	High-throughput Smart-seq method	Macosko EZ, et al. *Cell*. 2015 [33]
	2017	10x Chromium	yes	yes	PCR	3′ sequencing	+++	The most sensitive high-throughput scRNA-seq method.	Zheng GX, et al. *Nat Commun*. 2017 [67]
	2020	BAG-seq	yes	yes	PCR	3′ sequencing	++	Capturing nucleic acid directly in hydrogel	Li S, et al. *Genome Res*. 2020 [85]
Microwell-based high-throughput methods	2015	CytoSeq	yes	yes	PCR	3′ sequencing	++	Using microwell to isolate and label cells	Fan HC, et al. *Science.* 2015 [86]
	2015	Single-cell RNA printing	yes	no	IVT	3′ sequencing	++	Solid-phase capture of RNA	Bose S, et al. *Genome Biol*. 2015 [87]
	2017	Seq-Well	yes	yes	PCR	3′ sequencing	++	Semi-permeable polycarbonate membrane and surface-functionalized PDMS array	Gierahn TM, et al. *Nat Methods*. 2017 [89]
	2018	microwell-seq	yes	yes	PCR	3′ sequencing	++	Cheap agarose microarray	Han X, et al. *Cell*. 2018 [88]
Combinatorial indexing-based high-throughput methods	2017	sci-RNA-seq	yes	yes	PCR	3′ sequencing	++	High-throughput and low cost	Cao J, et al. *Science*. 2017 [68]
	2018	Split-seq	yes	yes	PCR	3′ sequencing	+++	High-throughput and low cost	Rosenberg AB, et al. *Science*. 2018 [69]
Spatial transcriptomics	2016	LCM-seq	no	no	PCR	full length	+	Providing spatial information	Nichterwitz S, et al. *Nat Commun*. 2016 [41]

* Number of cells analyzed in one experiment. +, Below 100; ++, between 1000–10,000; +++, above 10,000.

## Data Availability

Not applicable.

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
