# Peer review of "Microfluidics Facilitates the Development of Single-Cell RNA Sequencing"

_biosensors, 2022, doi:10.3390/bios12070450_

Round 1

Reviewer 1 Report

Dear Editor,

The manuscript entitled “Microfluidics Facilitates the Development of Single-Cell RNA Sequencing” by Yating Pan et al. presents in a comprehensive way the history of development and important technical points of scRNA-seq. The authors focus on microfluidics role in scRNA-seq technology development, mainly presenting droplet and microwell based formats. The authors also discuss the future directions for scRNA-seq, including perspectives on multi-omics and spatial analysis.

In my opinion, the manuscripts’ objective and perspective are very interesting, the manuscript is very well structured and well-written. The inclusion of figures is very helpful to the reader and the objective of the present manuscript is distinct, filling a gap in microfluidic technology for scRNA-seq. Therefore, I think that the manuscript should be accepted for publication as it is.

Reviewer 2 Report

The current manuscript entitled “Microfluidics Facilitates the Development of Single-Cell RNA Sequencing” by “Pan et al” reviewed and assessed the history of development and important technical point of scRNA-seq. Furthermore, focused on the role of microfluidics in facilitating the development of scRNA-seq technology. The manuscript seems good, interesting, and written well. The article can be accepted after addressing the following comments.

1.       In the introduction section provide some basic information on the Microfluidics.

2.       Some more protentional information should be added in the introduction section, seems some information is missing.

3.       How can the sensitivity (section 3.2.) be improved using Microfluidics.

4.       Section 6 caption is confusing, its “perspective” or “future perspectives”

5.       “Conclusion” should be revised as “Conclusions”

6.       Improve the quality of figure 2.
